# Microballoon Interventions for Liver Tumors: Review of Literature and Future Perspectives

**DOI:** 10.3390/jcm11185334

**Published:** 2022-09-11

**Authors:** Pierleone Lucatelli, Bianca Rocco, Simone Ciaglia, Elio Damato, Cristina Mosconi, Renato Argirò, Carlo Catalano

**Affiliations:** 1Vascular and Interventional Radiology Unit, Department of Radiological, Oncological, and Anatomo-Pathological Sciences, Sapienza University of Rome, 00161 Roma, Italy; 2Department of Radiology, IRCCS Azienda Ospedaliero-Universitaria di Bologna, Via Albertoni 15, 40138 Bologna, Italy; 3Diagnostic Imaging and Interventional Radiology, University Hospital of Rome Tor Vergata, 00133 Rome, Italy

**Keywords:** microballoon intervention, hepatocellular carcinoma, liver malignancies, balloon-occluded transarterial chemoembolization, balloon-occluded SIRT, balloon-occluded TACE and ablation

## Abstract

Background: Microballoon interventions (MBIs) have been proposed as useful tools to improve the efficacy of locoregional liver treatments. The aim of this systematic review was to summarize the existing evidence on procedural characteristics, safety, and efficacy of MBIs. Methods: PubMed and Cochrane Central Register of Controlled Trials were queried for original research articles evaluating MBIs in patients with liver malignancies from 2012 to August 2022. Search terms employed were liver malignancies, hepatocellular carcinoma, cholangiocarcinoma, liver metastases, microballoon transarterial chemoembolization, balloon-occluded trans-arterial chemoembolization, balloon-occluded selective internal radiation therapies, balloon-occluded TACE and ablation, and safety or oncological results or efficacy. Merely technical studies and animal studies were excluded. Results: Thirty-four original research studies and one abstract involving 744 patients treated with MBIs were included; 76% of the studies were retrospective, with low risk of bias and moderate-to-poor levels of evidence. Heterogeneity precluded meta-analysis. All studies proved MBI safety, which was not inferior to non-occlusive procedures. Balloon employment ameliorates oncological results, improving time to recurrence, objective response rate, and lowers need for retreatment. Conclusions: MBIs appear to be potential game changers in the treatment of liver malignancies. Multicentric, prospective and randomized studies are necessary to confirm these findings.

## 1. Introduction

Microballoon interventions (MBIs) consist in the execution of an embolization procedure immediately after the temporary occlusion of a vascular territory. The advantages of this novel technique lie in the possibility of preventing reflux of embolic material into non-target territories as well as in giving the operator the capability to perform a pressure-gradient driven embolization, which could not be otherwise achieved.

The underlying mechanism of action which permits this technical advantage is the following: the opening of the intersegmental arterial arcades determines a restoration of blood flow beyond the occluded segment, restoring hepatopetal flow towards lower-resistance areas such as the tumoral hypertrophic vasculature. Recent literature has been focused on the flow and concomitant pressure alterations associated with several different antireflux devices [1,2].

To date, this novel technique, introduced by Irie in 2012 [3], has been applied to the treatment of both primary and secondary liver tumors, as well as to different modalities of transarterial therapies: conventional and drug-eluting embolic transarterial chemoembolization (c-TACE and DEE-TACE, balloon-occluded-c-TACE -b-c-TACE- and balloon-occluded-DEE-TACE -b-DEE-TACE) as well as selective internal radiation therapies (SIRT and b-SIRT). In specific indications and clinical scenarios, microballoon TACE has been applied in combination with percutaneous ablation.

In the management of primary liver cancer, the role of transarterial therapies, both TACE and SIRT, is well established for very early, early and intermediate stage hepatocellular carcinoma (HCC) [4], while radioembolization is an option, according to ESMO guidelines [5], also for patients with advanced intrahepatic cholangiocarcinoma. Finally, with regard to metastatic disease, transarterial therapies are suggested as an option in those patients who are not candidates for curative treatments such as resection or thermal ablation [6,7,8].

Therefore, we performed a systematic literature review of the published evidence regarding use of MBIs for the treatment of liver malignancies, focusing on technical aspects, safety and efficacy in terms of oncological results.

## 2. Materials and Methods

### 2.1. Review Design, Search Strategy and Study Eligibility

This systematic review was conducted according to Preferred Reporting Items for Systematic Reviews and Meta-Analyses guidelines [9].

PubMed and Cochrane Central Register of Controlled Trials were queried for published literature in order to evaluate microballoon catheter interventions in patients with liver malignancies from 2012 to present. Reference lists of included studies were manually searched for other studies that potentially could meet the inclusion criteria.

To identify potential articles, the following search terms were used according to the PICO framework [10]: liver malignancies OR hepatocellular carcinoma OR HCC OR cholangiocarcinoma OR liver metastases; microballoon transarterial chemoembolization OR balloon-occluded transa-arterial chemoembolization OR b-TACE OR balloon-occluded selective internal radiation therapies OR b-SIRT OR balloon-occluded TACE and ablation; safety OR oncological results OR efficacy.

Studies were considered eligible if they included patients who underwent MBI to treat liver malignancies (histopathological diagnosis, with the exception of HCC in which accepted noninvasive criteria are allowed, according to Liver Imaging Reporting and Data System, LI-RADS).

All types of articles published in a peer-reviewed journal reporting efficacy or safety data were evaluated. Data from reviews were not extracted; however, reference lists were used as well as those from guidelines and commentaries. In the case of articles with overlapping cohorts, the one with the largest sample size or longest follow-up duration was included. All studies in which MBI was compared to an active comparator were included.

Exclusion criteria were the following: no employment of microballoon catheter; inadequate information regarding efficacy or safety of MBI, dose or name of the chemotherapeutic agent; embolization material not specified; article not written or published in English; tumor treatment not as primary indication to the employment of microballoon. Animal and non-clinical studies were also excluded.

Primary outcomes included technical aspects, safety (adverse events) and efficacy. Efficacy, in terms of oncological results, was evaluated according to modified Response Evaluation Criteria in Solid Tumors [11] (mRECIST) and Response Evaluation Criteria in Cancer of the Liver [12] (RECICL) for HCC; RECIST 1.1 [13] was used for metastases as well as for volumetric evaluation of tumor debulking. Where available, OS and time to recurrence were recorded. Other secondary clinical outcomes were included where available.

### 2.2. Study Selection and Data Extraction

Two reviewers (PL and BR) independently screened titles, abstracts, material and methods and results sections of each article in order to select eligible articles. In case of discordance, a third reviewer provided the final decision. Study duplicates were removed. Data on study design, patients’ characteristics, procedural details (type of microballoon employed, embolization technique, selectivity of embolization, embolic agent and chemotherapeutic drugs employed, changes in hepatic hemodynamics), adverse events and clinical outcome were extracted.

### 2.3. Data Analysis and Evaluation of Quality of Evidence

A quantitative meta-analysis was not applicable due to the heterogeneity in patient populations, procedural characteristics, embolic materials used, and outcome measurements. Results were summarized qualitatively by type of MBI.

Evaluation of the quality of evidence was performed according to criteria laid down by the Oxford Centre for Evidence-Based Medicine. Because the aim of this review was to identify all data on the value of microballoon applied to MBIs, no research studies were excluded on the basis of quality.

## 3. Results

The primary search resulted in 61 papers. After review, 34 original articles and one abstract were considered for data extraction regarding b-TACE, b-SIRT or combined treatment of b-TACE and ablation. Twenty-three papers were from Japan, where the use of microballoon catheters for selective TACE was first introduced.

Nine papers were reviews, commentaries or guidelines. These were excluded after review of their reference lists; other excluded papers did not employ microcatheters at all (*n* = 7) or not for tumor control (*n* = 2), were not in English (*n* = 2), had overlapping populations (*n* = 3), were duplicate studies (*n* = 1) or did not involve humans (*n* = 4). Quality of evidence was low due to the absence of randomized, controlled trials and the mostly retrospective nature of the papers (25/33, 76%).

All procedures were performed by an interventional radiologist. Results are displayed by the subtype of MBI (b-TACE, b-SIRT, b-TACE + MWA) and categorized based on technical aspects, safety and efficacy.

### 3.1. b-TACE

Thirty papers discussing b-TACE were selected. Four studies described two novel subtypes of b-TACE that differ from the standard procedure described for the first time by Irie [1]. These novel subtypes are selective occlusion of feeding arteries (SOFA-TACE) [14] and repeated alternate infusion of cisplatin solution and gelatin slurry distal to balloon occlusion (RAIB-TACE) [15,16,17]. The results of these four studies are reported separately.

In all studies, HCC tumors were treated and b-TACE was performed through femoral access and 4 or 5 Fr sheath.

#### 3.1.1. Procedural Characteristics

All papers reported the same principles of b-TACE: once the microcatheter is advanced as selectively as possible, it is inflated in the target vessel to a diameter 5–10% larger than that of the occluded artery. Microcatheter diameters employed ranged from 1.8 Fr to 3 Fr (Attendant, Terumo, Tokyo, Japan; Logos, Piolax, Kanagawa, Japan; Occlusafe, Terumo; Optimo PB; Tokai Medical Products; Sniper, Embolx, Sunnyvale, California). Some authors reported additional selective microcatheterism and embolization if, after embolization under microballoon occlusion, other feeders were detectable [18].
Several studies focused on the demonstration of flow redistribution (Table 1) that occurs after microballoon inflation. Because this mechanism cannot be directly quantified, several different surrogate indicators were investigated. In c-TACE, the concentration of chemotherapeutic drugs can be indirectly deduced by evaluating tumor opacification determined by lipiodol emulsion (LE) deposition on plane CT or cone beam CT (CBCT) [19]. Whereas another group demonstrated in vivo, with unenhanced CBCT, how DEE-b-TACE was able to ameliorate the concentration of chemotherapeutic drug carried by drug-eluting embolics within the tumor [20].

Irie, the pioneer of this technique, introduced the concept of “balloon-occluded arterial stump pressure (BOASP)” in 2012; this consists in the assessment of arterial pressure at the tip of the inflated microcatheter. A BOASP of at least 64 mmHg was found to correlate with a complete opacification of the tumor, with no deposition of LE in the non-target surrounding liver parenchyma. In patients in which an optimal deposition of LE was not obtained, BOASP was significantly higher (92.3 ± 7.4 mmHg, range 83–100 mmHg, *p* = 0.00004).

Matsumoto et al. [21] evaluated BOASP at each hepatic arterial level before performing c-b-TACE. BOASP in “non-targeted” arteries (lobar artery) was significantly greater than in ‘‘selective’’ arteries (subsegmental or segmental) (*p* = 0.0147). BOASP in A1, 4, 8 and the anterior segmental arteries were significantly greater than in the other subsegmental and segmental arteries (*p* = 0.0007), suggesting that at least segmental catheterization should be performed in order to obtain flow redistribution. Kakuta et al. [22] demonstrated that flow redistribution occurs immediately after balloon occlusion, since no significant differences were found in BOASP values measured immediately after and 5 min after balloon occlusion (*p* = 0.124). Hemodynamic changes were investigated also with contrast enhanced ultrasound (CEUS) in a case series of two patients [23] and with CT or CBCT post contrast acquisition with balloon deflated and inflated [24,25,26,27]. Inoue in 2019 [27] proposed the employment of a double balloon (a 5.2 Fr balloon inflated in proper hepatic artery and a selective 1.8 Fr in the tumor’s feeder) in order to ensure a flow redistribution in those patients in which the target BOASP of >64 mmHg is not achieved by the sole microballoon occlusion.

**Table 1 jcm-11-05334-t001:** Summaries of studies focused on demonstration of flow redistribution.

Study, Type of Research		Patients (*n*)	Age	M/F	HCC (*n*)	HCC Dimension (mm Mean ± SD or Range/Median)	Technique	Aim	Main Findings
Ishikawa 2017 [26]	retrospective	52	72.32 ± 7.78	40/12	52	27.69 ± 6.82	c-b-TACE	Evaluate hemodynamic changes with/without balloon occlusion of the hepatic artery, correlation of cone-beam CT (CBCT) pixels, and CT value after b-TACE.	After balloon occlusion, CBCT pixel values increase (*p* = 0.048). Intratumoral CT values after b-TACE were lower with decreased CBCT pixel values than with increased CBCT pixel values.
Ishikawa 2016 [19]	retrospective	82	71.4 ± 7	65/17	82	31.3 ± 5.8	c-b-TACE	Whether Lipiodol tumoral enhancement on plane cone-beam CT (CBCT) can be used to predict tumor response as CT scan.	Significant correlation between plain CT value and CBCT value, with a Pearson correlation coefficient of 0.912 (*p* < 0.001).
Sugimoto 2014 [23]	case report	2	81 and 64	1/1	2	30 and n/a	c-b-TACE	Depicting hemodynamic changes with CEUS after microballoon occlusion.	Hemodynamic changes during B-TACE were depicted and evaluated by CEUS.
Matsumoto 2015 [21]	retrospective	47	74 ± 11	33/14	n/a	31 ± 19	c-b-TACE	Evaluate BOASP at each hepatic artery segment (non targeted: lobar vs. targeted: segmental and subsegmental) before b-TACE.	‘‘Non-targeted’’ BOASP was significantly greater than ‘‘selective’’ BOASP (*p* = 0.0147), that should be preferred for efficient b-TACE. BOASP in A1, 4, 8 and the anterior segmental arteries were significantly greater than in the other subsegmental and segmental arteries (*p* = 0.0007), suggesting a potential less efficacy of b-TACE on HCC localized in those segments.
Yoshimatsu 2016 [25]	retrospective	24	73 ± 7.5	13/11	27	20.3 (10.2–47.3)	c-b-TACE	Evaluate changes on CT during hepatic arteriography (CTHA) and during arterial portography by balloon occlusion of the feeder artery and their relationship with LE accumulation in the tumor.	Tumor enhancement on selective CTHA frequently changed after balloon occlusion, which did not correspond to accumulated iodized oil in most cases.
Kakuta 2015 [22]	case series	27	68.3 (42–88)	15/12	219	76.5 (10–486)	c-b-TACE	Analyze temporal variations in stump pressure and influencing factors.	No significant difference BOASP between immediately after and 5 min after balloon occlusion (*p* = 0.124). Following intra-arterial injection, mean arterial blood showed a significant increase of 21.5 mmHg from the value immediately after balloon occlusion (*p* < 0.0001, Student’s *t*-test). Mean arterial blood pressure after balloon deflation following intra-arterial injection was not significantly different from that before balloon occlusion. Contrast to Noise Ratio is significantly higher than those before balloon occlusion.
Irie 2012 [1]	prospective	42	72.2 ± 7.9	32/10	43	38.7 ± 23.2	c-b-TACE	reveal the mechanism of dense accumulation of lipiodol emulsion (LE).	The BOASP in group 1 (patients in which LE after filling the peritumoral vessels continued to fill the tumor and not the parenchyma) was 33.8 ± 12.8 mmHg (range 13–64 mmHg) and in group 2 (LE in nontumorus parenchyma) was 92.3 ± 7.4 mmHg (range 83–100 mmHg) (*p* = 0.00004, Welch’s *t* test). The LECHL ratio in group 1 was 18.3 ± 13.9 (range 2.9–54.2) and that in group 2 was 2.6 ± 1.1 (range 1.7–4.2). There was a statistically significant difference in the LE concentration ratio of HCC to embolized liver parenchyma between the groups (*p* = 0.000034, Welch’s *t* test).
Asayama 2016 [24]	retrospective	29	73.1 ± 2.1	21/8	35	16.6 (9–40)	c-b-TACE	Predicting therapeutic effects on the base of CT angiography performed from deflated and inflated balloon.	When injecting from inflated balloon the tumors with filling defect (group C) showed significantly poor TE compared to tumor without corona enhancement (group B) (*p* = 0.002). Tumors without corona enhancement (Group A) and group C differed but not significantly (*p* = 0.075). There was no significant difference between Group A and Group B (*p* = 0.350). CT values of the lesions were correlated with the TE (*p* = 0.037). Group C as a significant factor associated with the worst short term TE bearing an odds ratio of 8.34 (95% confidence interval 1.49–68.8).
Lucatelli 2022 [20]	retrospective	27	n/a	n/a	27	27 (CI 95%: 23.0–35.1)	DEE-b-TACE	To evaluate in vivo the role of the micro-balloon by comparing microspheres deposition in DEE and DEE-b-TACE.	Contrast, signal-to-noise ratio, and contrast-to-noise ratio were all significantly higher in DEE-b-TACE subgroup than DEE-TACE (*p* < 0.05). Histological explanted liver analysis, trend for higher intra-tumoral localization of embolic microspheres for DEE-b-TACE in comparison with DEE-TACE.
Inoue 2019 [27]	prospective	9	69.4	7/2	9	23.3 ± 22.18	c-b-TACE	Assess the change in hepatic arterial blood pressure (HABP) and computed tomography during hepatic arteriography (CTHA) using a balloon inflated in the hepatic artery and a microballoon catheter selectively.	Double balloon technique allows to achieve less BOASP and better LE concentration. Occlusion of the PHA using the double balloon technique is worth attempting when HABP is >64 mm Hg by microballoon occlusion prior to b-TACE. CTHA using a double balloon catheter could assess hemodynamic changes via collateral arterial blood flow by balloon occlusion of the intrahepatic and extrahepatic arteries.

Chemoembolization techniques reported were predominantly c- TACE, in accordance with the Eastern Asian provenience of the studies. In fact, in 23 out of 26 papers, b-TACE was performed with chemotherapeutic drug emulsion in Lipiodol (c-b-TACE), followed by embolics administration (gelatine sponge 1–2 mm in all cases, except Goldman et al. [28], who employed microspheres Embozene 100 μm). Three European studies [20,29,30] described the application of b-TACE with drug eluting embolics veiculating chemotherapeutic drug (DEE-b-TACE), while a European [31] and an American study [28] reported the application of b-TACE to both of these techniques. The diameter of microparticles employed in DEE-b-TACE ranged from 75 μm to 300 μm.Several chemotherapeutic regimens are reported in b-TACE procedures. The reported LE techniques and dosage regimens did not differ from standard c-TACE. Regarding c-b-TACE, 11 (11/23, 47.8%) studies employed exclusively miriplatin [15,18,19,23,24,26,27,32,33,34,35], 3/23 (13%) epirubicin [31,36,37], 1/23 (4.3%) doxorubicin [28] and 1/23 (4.3%) cisplatin [38]. Six papers reported cases with multiple drugs [3,22,25,36,39,40]. In DEE-b-TACE, epirubicin was the sole drug employed in three papers [20,29,31] and doxorubicin in two [28,30].

#### 3.1.2. Safety Profile and Efficacy

Safety and efficacy data were extracted from 16 papers (640 patients, 426:204 males:females). Safety was evaluated according to CTCAE v4–5 [41] or CIRSE classification of complications [42], with efficacy with modified Response Evaluation Criteria in Solid Tumors or, in the majority of East Asian studies, with the Response Evaluation Criteria in Cancer of the Liver (RECICL) (Table 2).

Regarding safety data, the majority of complications reported, such as post embolization syndrome (PES) and elevation of transaminase levels, were related to the TACE procedure per se, while the only adverse event strictly related to the employment of the microballoon catheter was the dilatation of the artery segment where the balloon was inflated. This complication was independently reported by two authors [31,40] as a collateral finding at imaging follow up, with an incidence of 1.1% and 2.8%, respectively. Comparative studies including TACE without balloon occlusion (both c-TACE and DEE- TACE) demonstrated no significant statistical differences in severe adverse events (according to CTCAE: greater than Grade 3), although significant statistical differences in increase of serum AST, ALT, ALP and white blood cell count were independently reported by two groups [33,43]. All reported modifications of serum values and hepatic function returned to normality within maximum 1 month.

The largest cohort of patients treated with DEE-b-TACE was a retrospective multicenter European study led by Golfieri [31] (69 patients). Complications reported were all grade 1–2 (CTCAE v5) and consisted of PES (41.8%), asymptomatic abscess (2.2%) and hepatic pseudoaneurysm (1.1%). All complications were treated conservatively and no significant statistical differences were found when compared to a c-TACE cohort. Hatanaka et al. [43], in the retrospective study that involved the largest number of patients undergoing c-b-TACE (66 patients), reported Grade 3 elevation of total bilirubin (6.1%) and ALT (9.1%), leukocytopenia (12.1%) and thrombocytopenia (7.6%), all managed with conservative therapy. A biloma (1.5%) that required drainage occurred in this case series. The occurrence of biliary severe complications was correlated to common biliary duct by Maruyama et al. [37].

In regards to efficacy, MBIs were shown to ameliorate the oncological performance of TACE both with its application in the conventional approach and in DEE-TACE. c-b-TACE, compared to c-TACE, demonstrated a better LE deposition when the occlusion occurred at the level of subsegmental arteries [37]. A better therapeutic effect (*p* = 0.016) and improved control rates of the primary nodule (*p* = 0.0016) were proved in a study by Irie et al. [18]. The same study showed no statistically significant differences in overall survival or tumor-free rates in the liver.

The balloon employment demonstrated, in a study led by Golfieri [31], that it could lead to the achievement of a higher complete response rate in the treatment of 30–50 mm HCC (*p* = 0.047) as well as a significantly lower re-treatment rate after a single TACE in comparison with c-TACE (12.1 vs. 26.9%, respectively; *p* = 0.005).

Only one study [29] reported two cohorts of patients treated with DEE-TACE with and without microballoon (respectively 22 vs. 53 patients). Populations were homogeneous except for tumor dimension, which was significantly larger in DEE-b-TACE (27 mm [CI 95%: 23.0–35.1] vs. 15.5 mm [CI 95%: 14–22.5], *p* = 0.005). In this series, oncological responses evaluated using mRECIST criteria were similar at all time points for the two treatments, with the exception of objective response rate at 9–12 months, where DEE-b-TACE showed a trend of better oncological response and longer time to recurrence over DEE-TACE in patients presenting with larger tumors. This study was the one presenting the longest oncological follow-up with complete response, partial response and progressive disease rate at 9–12 months of 68.4%, 10.5% and 21.1%, respectively.

Overall survival rates ranged from 89.6% and 85.7% at 1 year, from 57.3 and 52.3% at 2 years and from 46.7 to 17.1% at 3 years [38,43,44].

#### 3.1.3. Variations of b-TACE

##### Selective Occlusion of Feeding Arteries (SOFA) TACE

The SOFA-TACE technique, described by Yu in a prospective Chinese study published in 2022 [14], consists of multiple tumor feeder catheterization. In particular the dominant feeder is catheterized for delivery of the chemotherapeutic agent (2.4 Fr Merit Maestro, Merit Medical Systems, South Jordan, UT, USA), while another arterial feeder or a common trunk of all other arterial feeders is selectively catheterized with a microballoon catheter (Occlusafe, Terumo Clinical Supply, Gifu, Japan). In this technique, the drug is delivered through the non-occludent microcatheter, which is inflated in order to temporarily arrest the flow in all the other tumor feeders and consequently reduce the number of feeding vessels that need to be catheterized. The study involved eight patients, [six males, median age 64.5 years old (interquartile range (IQR) 60–68.8 years), median largest tumor dimension was 47 mm (IQR 32–61 mm)]. Number of feeders ranged from two to five and the whole tumor vasculature was completely filled up through delivery at one arterial feeder in seven of eight cases (87.5%). Serum parameters were altered by 1–2 grades (CTCAE v5) and returned to normality within one month. A sustained complete response was achieved in all cases up to a median surveillance period of 25 months (range 22–28 months).

##### Repeated Alternate Infusion of Cisplatin Solution and Gelatin Slurry Distal to Balloon Occlusion (RAIB) TACE

In 2019, Irie [15] modified the classical c-b-TACE technique in the RAIB-TACE, crushing the 1 mm particles (Gelpart, Nipponkayaku, Tokyo) into gelatin slurry of smaller fragments (130–200 micron in length) and alternating its infusion to cisplatin powder dissolved in warmed saline as 1 mg/mL. This technique required a more proximal catheterization in comparison to b-TACE. The same group investigated the safety and efficacy of this approach in the treatment of small HCC adjacent to the Glisson sheath in HCC larger than 70 mm [16] and in a Phase I/II Multicenter Prospective Study of Safety and Efficacy [17]. Objective response ratio (CR or PR = 100%) of nodules adjacent to the Glisson sheath treated with RAIB-TACE was significantly higher than that in c-TACE group (OR = 62.1%) (*p* = 0.008, Fisher’s test). The objective response rate of 100% at 1–3 months’ follow-up was obtained also in HCC > 70 mm, whereas in this population a hepatic abscess occurred due to the extensive necrosis obtained, requiring drainage positioning.

**Table 2 jcm-11-05334-t002:** Summaries of studies focused on safety and efficacy of MBI.

Study	Type of Research	Patients	Age	M/F	HCC Nodules (n)	HCC (mm, Mean ± SD or Range/Median)	Technique	Aim	Safety	Efficacy	Main Findings
Golfieri 2021 [31]	retrospective	22	68 (40–91)	18/4	179	36 (9–159)	c-b-TACE	Evaluate TACE performance with and without balloon occlusion, assess in which size range offer higher CR/OR in a single session.	AEs similar, PES 41.8%, asymptomatic abscess (2.2%) hepatic pseudoaneurysm (1.1%) All complications grade 1–2 (CTCAE v5).	1 month FU: CR 68.2%, PR 27.3%, SD 0, PD 4.5%	in 30–50 mm HCC, B-TACE achieves higher CR rates (*p* = 0.047), whereas in smaller nodules (<30 mm), cTACE can suffice in achieving a good CR rate. The statistically significant lower re-treatment rate of the B-TACE cohort after a single procedure reduced the risk of complications due to multiple TACE, which could worsen the patient prognosis (12.1 vs. 26.9%, respectively; *p* = 0.005).
		69		57/12			DEE-b-TACE			1 month FU: CR 56.5%, PR 31.9%, SD 7.2%, PD 4.3%	
Lucatelli 2021 [29]	retrospective	22	65.9 ± 13.8	19/3	35	27 [CI 95%: 21.6–32.4] median	DEE-b-TACE	Safety and efficacy DEE-bTACE vs. DEE-TACE	PES 36.4%. (CTCAE v5) grade 3: 4.5%, grade 2: 18.1%. No statistical differences with DEM-TACE.	1 month CR 40.0%, PR 25.7%, SD 25.7%, PD 8.6%/3–6 months FU CR 44.8%, PR 27.6%, SD 20.7%, PD 6.9%/9–12 months FU CR 68.4% PR 10.5%, SD 0, PD 21.1%	mRECIST oncological response at all time points (1, 3–6 and 9–12 months) for both treatments were similar, with the exception of Objective response rate at 9–12 months. b-TACE showed a trend of better oncological response over DEM-TACE with and longer TTR with a similar adverse events rate, in patients presenting with larger tumors.
Bucalau 2020 [30]	prospective	24	66 ± 10.1	23/24	40	32.7 ± 11.8	DEE-b-TACE	DEE-bTACE safety and efficacy	Clinical grade 1/2 toxicities in 25.7% abdominal pain (17.1%).	1 month FU: CR: 41.2%, PR 29.4%, SD 29.4%, PD 0%	Safe and effective.
Pyeong Hwa Kim 2020 [38]	retrospective	60	61.4 ± 10.0	49/11	60	30 ± 25	c-b-TACE	c-bTACE efficacy in the management of residual or recurrence of HCC previously treated with cTACE.	PES 90%, Acute Kidney injury 1.6%, Asymptomatic ischemic cholangiopathy 1.6%, partial liver infarction 1.6%. Increase AST and bilirubin.	1–3 months FU: CR 25%, PR 75%. OS at 1 year: 89.6%.	safe and effective for the treatment of HCC refractory to C-TACE. BCLC stage C and multiplicity of HCC were independent factors associated with TTP after B-TACE.
Minami 2015 [36]	retrospective	17	74.4 ± 6.2	13/4	32	20 ± 9	c-b-TACE	c-b-TACE dense LE accumulation and efficacy in patients with countable and uncountable HCC	Grade 2 or grade 1 adverse events: increased ALT 18.5% All these events resolved within 2 weeks.	1–3 months FU: TE4 43.8%, TE3 12.5%, TE2 37.5%, TE1 6.3%	b-TACE did not reduce the efficacy of retreatment for HCC with an insufficient outcome from conventional TACE, but it could not improve the efficacy of treatment for uncountable multiple HCCs.
		10	75.3 ± 6.3	7/3	n/a	approx 10–20 mm				1–3 months FU: CR and PR 0% SD 10%, PD 90%	
Ogawa 2016 [33]	retrospective	33	74 (41–88)	19/14	62	22 (7–90) median	c-b-TACE	Safety and efficacy of c-bTACE vs. cTACE using miriplatin.	Increase of ALP e WBC significantly higher in bTACE. All returned to baseline. Common fever and nausea.	1–3 months FU: TE4 49.2%, TE3 + 2 + 1 50.8%	complete coverage of the lesion with LP 67.7% of cases in the B-TACE group and 59.0% in the C-TACE group, with no significant difference between the groups (*p* = 0.370). Local efficacy was significantly higher in nodules treated by B-TACE.
Matsumoto 2015 [40]	retrospective	31, (70 c-bTACE)	73 ± 7.5 (56–85)	20/11	n/a	<30 mm: 18 (58%), 30–50 mm: 8 (26%), >50 mm: 5 (16%)	c-b-TACE	Safety and technical success.	PES, pseudoaneurysm 2.8%.	n/a	B-TACE using the 1.8-Fr tip microballoon catheter is a safe procedure.
Arai 2014 [34]	retrospective	49	71.9 (62–84) median	33/16	49	29 (8–73) median	c-b-TACE	Safety and efficacy of c-b-TACE vs. c-TACE using miriplatin.	ALT significantly higher in B-TACE (*p* < 0.05), to baseline within 1 month.	1 month FU: TE4 55.1%, TE3 38.8%, TE2 4.1%, TE1 2%	Significantly higher mean miriplatin total dose (*p* < 0.01) and TE values (*p* < 0.05) in the B-TACE group.
Irie 2016 [18]	retrospective	28	72.5 ± 9	22/6	36	39.2 ± 22.9	c-b-TACE	Safety and efficacy of c-b-TACE vs. c-TACE in naive patients with one or two HCC.	No severe TACE-related complications.	cTACE vs. bTACE: control rates of primary nodule (Hazard ratio (95% CI) 3.92 (1.64–9.37), *p* = 0.002), overall survival rates (1.87 (1.02–3.42) *p* = 0.04)	Better treatment effect in the B-TACE group (*p* = 0.016); Control rates of the primary nodule improved in B-TACE (*p* = 0.0016). No statistically significant differences in overall survival or tumor-free rates in the liver. B-TACE independent factor to improve control rates of the primary nodule (*p* = 0.002).
Ishikawa 2014 [35]	prospective	51	70.9 ± 9.17	35	55	less than 50 mm	c-b-TACE	Evaluate predictive factors of local recurrence after c-bTACE with miriplatin.	78.4% PES, anorexia 31.3%. AST and ALT elevation in all but returned to normality within 2 weeks.	Mean value CT post: 325.7 HU, overall recurrence rate 11.1%	Local recurrence rate significantly different in the higher-than-mean CT value group (6 months 4.8%, 16.0% at 12 month) than in the lower-than-mean CT value group (6 months, 15.2% and 12 months 32.9%). CT value after B-TACE correlated with local recurrence (hazard ratio 0.11; 95% confidence interval 0.01–0.98; *p* = 0.048).
Maruyama, 2015 [37]	retrospective	50	n/a	n/a	50	3.2 ± 2.8	c-b-TACE	Accumulation of lipiodol emulsion (LE) and adverse event compared to c-TACE.	Elevation of AST and ALT (P0,05). liver abscess 6% and liver infarction (2%). Patients with biliary severe complications had common bile duct dilatation.	Mean LE ratio of the B-TACE at the level of subsegmental: 8.24 (6.88–8.34) vs. C-TACE 4.18 (3.57–4.80) (*t* test: *p*\0.05).	bTACE is safe and can cause severe complications in patients with common bile duct dilatation. bTACE can obtain a better LE ratio when performed subsegmental.
Kawamura, 2016 [32]	retrospective	30	76 (54–88)	13	47	20 (6–55)	c-b-TACE	Efficacy and predictive factors in c-b-TACE performed with miriplatin in patients with 4 or less HCC.	Delayed diagnosis of postembolization syndrome required a re-admission. No others complications than PES.	TE4 in 51%, TE3 in 9%, TE2in 19%, TE1 in 21% (OR in 60% of nodules). With appreciable portal vein during bTACE: TE4 in 88%, TE3 in 0%, TE2 in 0%, TE1 in 12% (OR in 88%)	Independent factors for OR: portal vein visualization during B-TACE (hazard ratio (HR), 15.74; 95% CI, 1.78–139.15; *p* = 0.013); tumor on the subcapsular portion (HR, 8.30; 95% CI, 1.37–50.36; *p* = 0.021); and successful subsegmental artery embolization (HR, 5.95; 95% CI, 1.17–30.33; *p* = 0.032)
Shirono, 2018 [44]	retrospective	35	73 (61–85)	21	40	21 (12.25–65)	c-b-TACE	Efficacy of c-b-TACE performed with epirubicin vs. miriplatin.	AE > grade 3 (CTCAE v4): elevation of transaminase (28.57%), liver dysfunction (2.8%), obstructive cholangitis (2.8%). Symptoms improved with conservative treatments.	TE4 52.5%, TE3 15%, TE2 25%, TE1 7.5%. OS 1 year 85.7%, 2 years 52.3%, 3 years 17.1%. TE4 rate of each regimen (i.e., epirubicin and miriplatin) was 64% and 33% respectively. TTP: epirubicin 15.1 months, miriplatin 3.2 months	Epirubicin had a positive tendency in TE4 rate compared with miriplatin (*p* = 0.058) and significantly prolonged the local TTP of the targeted lesions (*p* = 0.0293).
Goldman2019 [28]	retrospective	13	65 ± 7	11	15	27 (11–59)	c-b-TACE + DEE-b-TACE	Safety and efficacy b-TACE.	Serum chemistry analyses no significant difference	6 weeks: CR 60%, PR 33.3%, no SD or PD	Safe and effective.
Hatanaka 2017 [43]	retrospective	66	75 (IQR; 68.3–79)	45	n/a	25.5 (IQR; 18–37) mm.	c-b-TACE	Predict overall response and overall survival.	Grade 3 elevation of total bilirubin 6.1%, ALT 9.1%, leukocytopenia 12.1% and thrombocytopenia 7.6% (conservative therapy). Biloma 1.5% (percutaneous transhepatic biliary drainage).	CR 53%, PR 10.6%, SD 19.7%, 16.7%. OS 1 year 76.8% (95% CI: 64.5–85.3%), 2 years 57.3% (95% CI: 42.3–69.7%), 3 years 46.7% (95% CI: 30.7–61.2%)	Number of tumors (hazard ratio (HR) 4.44; 95% confidence interval (CI) 1.26–15.7; *p* = 0.021) and α-fetoprotein level (AFP; HR 11.40; 95% CI 2.75–46.9; *p* < 0.001) were significantly associated with the tumor response. Albumin (≥3.4 g/dL) (HR 0.28; 95% CI 0.12–0.63; *p* = 0.002) and overall response (CR + PR) (HR 0.33; 95% CI 0.16–0.71; *p* = 0.004) were significantly associated with the OS.
Shirono 2022 [39]	retrospective	30	74 (62–88)	21	33	21.0 (11.3–65)	c-b-TACE	Maintaining a durable CR after c-TACE, DEE-TACE or c-b-TACE.	AE > grade 3 (CTCAE version 5.0) 36.6%. (conservative treatment).	Local recurrence free (LRF) 1180 days; TTR 39.3 months, mean OS 41.4 months	B-TACE was an independent factor for the LRF period. B-TACE had a significantly longer LRF period than C-TACE and DEB-TACE.

Safety and efficacy of this newly developed technique was reported in a multicenter prospective study that involved for 43 patients with HCC from four medical centers [17]. Adverse events were facial swelling and skin rash (2.3%), dissection of the celiac artery (2.3%) and bland portal vein thrombus (2.3%). No major adverse events were identified and deterioration of Child–Pugh classification occurred in 5.3% of cases. The RAIB-TACE achieved a 73.2% OR rate (95% confidence interval [CI], 57.9–84.4%) and a CR rate of 22.0%.

### 3.2. b-SIRT

Microballoon applications in SIRT were reported in an abstract in 2017 [45] and in one study in 2022 [20].

The abstract by Saltarelli et al. [45] reported a series of nine patients with unresectable liver metastases who underwent SIRT with the employment of the microballoon to avoid extrahepatic non targeted implantation of Y90 microspheres, without coiling the gastro-duodenal artery. No extrahepatic non targeted implantation in the gastro-duodenal territories occurred.

The study conducted by Lucatelli et al. [20] compared 2D/3D dosimetry in single-photon emission computed tomography after SIRT versus b-SIRT. The only technical difference between SIRT and b-SIRT was the employment of a 2.7 Fr microcatheter (Occlusafe, Terumo Europe NV, Leuven, Belgium). Both procedures were performed using resin-based microspheres.

A post-therapy SPECT/CT scan was performed between 1 and 20 h after SIRT to evaluate the 90Y-microspheres distribution. Accuracy and intensity of 90Y-resin-microspheres activity distribution were evaluated by comparing the 2D activity intensity peak (pixel value) of the signal along a line crossing the treated area of patients treated with SIRT with and without microballoon. The 3D dosimetry permitted the evaluation of the effective dose (Gy) delivered to the target nodule and normal liver per unit cumulated activity (GBq).

b-SIRT demonstrated a better dosimetry profile both in 2D and 3D analyses. In 2D evaluation, the activity intensity peak was significantly higher in the b-SIRT subgroup than SIRT (987.5 ± 393.8 vs. 567.7 ± 302.2, *p* = 0.005), a higher quantity of Y90-microspheres was delivered to the target area of treatment with the same administered activity. In regard to the 3D dose analysis, the mean dose administered to the treated areas was significantly higher in the b-SIRT group than SIRT (151.6 Gy ± 53.2 vs. 100.1 Gy ± 43.4, *p* = 0.01) with almost no increase of the mean dose delivered to the normal liver (29.4 Gy ± 5.7 vs. 28.0 Gy ± 8.8, *p* = 0.70).

### 3.3. b-TACE + Ablation

The combination of MBI with radiofrequency thermal ablation (RFTA) in a single-step procedure was reported in 2015 by Iezzi et al. [7] in a prospective study that involved 40 patients with a single HCC (mean size 47 ± 11 mm). After microballoon inflation in the target vessel, RFA was performed and, post-ablation, drug eluting embolics were administered. No worsening of the Child–Pugh score, nor any complications related to the use of the balloon-occlusion technique were recorded. Complete response at one month was achieved in 80% tumors (maintained in 62.5%) and 20% obtained a PR.

The same group in 2017 [8] retrospectively compared the single-step DEE-b-TACE + RFA performed in 25 patients with compensated cirrhosis and single HCC > 30 mm (median size 45 mm; range, 30–68 mm) with a cohort of 29 patients who underwent liver resection (LR). While one death and one major complication (4%) were observed in the LR group, no major complications were reported in the DEE-b-TACE + RFA cohort. LR achieved lower tumor recurrence rates than DEE-b-TACE + RFA, but 3-year OS rates were not statistically different between the two groups.

In 2022, a multicentric study applied DEE-b-TACE to microwave ablation (DEE-b-TACE + MWA) in a single-step procedure [6]. This retrospective study involved 23 patients with liver malignancies >30 mm (both primary and secondary, maximum mean diameter of lesions 44 mm ± 10 mm). No complication occurred and CR and PR rates were, respectively, 91.3% and 8.7% at 1 month, 85.7% and 9.5% at 3–6 months. Progression of disease was 4.7% at 3–6 months due to extra-hepatic progression. Among partial responders, the average percentage of tumor volume debulking was 78.8% (±9.8%). Volume evaluation revealed a discrepancy with the expectations, based on vendor charts, with a median volumetric increase of the necrotic area of 103.2% (±99.8). In 95.7% of cases it was observed that the necrotic area showed a non-spherical shape corresponding to that of the vascular segment occluded during ablation, setting the ground for the novel concept of percutaneous thermal segmentectomy.

## 4. Discussion

MBI has emerged as an adjunctive tool in the field of liver embolization procedures, having already treated 744 patients worldwide. Balloon-occluded procedures have been demonstrated to be safe, with an adverse event rate shown to be equivalent to that of non-occlusive procedures. Further, it has been shown that they are able to positively impact the oncological outcome of treated patients, regardless of the embolic agent employed.

Regarding safety, none of the studies reported a significant difference in the rate of complications between procedures performed with and without balloon occlusion. The only reported complication directly related to the employment of microballoon was vascular dilatation at the site of balloon inflation. In all reported cases, this event was asymptomatic and was revealed during routine contrast enhanced CT follow-up and did not require further treatment.

Oncological response, despite being negatively influenced by different systems of evaluation across countries (RECICL and mRECIST), a difference that impeded a meta-analysis, demonstrates a better performance. The employment of balloon occlusion has been demonstrated to be an independent factor in maintaining a sustained complete response and to prolong the time to recurrence as well as being capable of reducing retreatment need and of obtaining higher complete response rate in nodules of 30–50 mm.

In spite of the fact that a variety of microcatheters produced by different vendors, as well as multiple techniques, have been reported in the literature, there is no evidence, to date, demonstrating that one is superior to the other. What clearly emerges from the literature, instead, is that MBIs could be considered to be an upgrade of interventional oncology liver embolization procedures, with the added advantage of being feasibly employed with embolics/radioisotopes which operators are likely already familiar with, thus not radically changing clinical practice but acting as a booster for clinical response.

Limitations of this systematic review are the quality of the studies included, most of them retrospective, the different techniques employed in these studies and the different outcomes measured. These aspects limited a global evaluation of the clinical impact of microballoon employment.

Finally, future research will have to address longer-term follow-up and wider cohorts in order to build robust evidence showing that patients can benefit from balloon-occluded TACE procedures. If the combination of MBI and ablation will be confirmed in ongoing prospective multicenter studies to be as promising as initially reported, this may be a game changer for patient management and liver cancer.

## Data Availability

All data can be found on PubMed.

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
