# Peer review of "Microballoon Interventions for Liver Tumors: Review of Literature and Future Perspectives"

_jcm, 2022, doi:10.3390/jcm11185334_

Round 1

Reviewer 1 Report

In this Systematic Review, the authors present the state of the art of microballoon interventions for hepatic malignancies. The authors have condensed a large amount of relevant information in an easy-to-follow review paper. I think the content should not be changed or completed, but the language-related issues should be carefully revised by the authors.

All my comments are minor comments:

- General: authors should choose between American English spelling or British English spelling and be consistent with it throughout the manuscript. For example, "haemodynamic" (line 164) vs. "hemodynamic" (line 101). In general, the language-related issues should be carefully reviewed by a native English speaker.

- Line 16: It should say "evidence" instead of "evidences".

- Line 19: The " should be closed after the last keyword.

- Line 24: It should say "research studies", instead of "researches".

- Line 25: 75.74% could be replaced by 76%. I think there is no need to be so precise.

- Line 38: The authors maybe meant "rely on", instead of "relieve in".

- General: Brackets should be used for citation [] and references should be written according to the indications provided by the journal (https://www.mdpi.com/journal/jcm/instructions#preparation)

- Line 92: the authors maybe meant "metastases"

- Line 94: I would join this sentence to the pevious paragraph.

- Line 101: It should say "hemodynamics" instead of "hemodynamic".

- Lines 110-111: It should say "research studies".

- Line 112: I think it should say "twenty-three".

- Line 112: dash (-) should be replaced by an em-dash (—).

- Line 113: "where" instead of "were".

- Line 113: I think the dash (-) before the full stop is not necessary.

- Lines 128-129: I would join this sentence to the previous paragraph.

- Lines 139, 176, 186, 200, 225: I think the keywords should not be underlined.

- Line 146: Did the authors mean "vehiculated", with the meaning of "carried"?

- Line 149: "balloon-occluded arterial stump pressure"

- Line 165: What is the meaning of CEUS?

- Table 1: Decimal separator must be a "." and not ",". Some headings are missing. First column could be "Study", second column could be "Type of research" or something. Ninth column could say "Objective" or "Aim". Last row, in findings, it should say "allows" instead of "allow".

- Line 181, 185: I guess the unit of the microsphere diameter is micrometer. It says now "um".

- Line 195: "datas" should be corrected to "data".

- Line 205: "founding" should be corrected to "finding".

- Line 226: Did the authors mean "and" instead of "than"?

- Line 246: Decimal separator should be ".", not ",".

- Table 2: Some headings are missing. First column could be "Study", second column could be "Type of research" or something. Ninth column could say "Objective" or "Aim".

- Line 336: What does LR stand for?

- Line 368: What is the best unit to measure nodules? mm or cm? In the text you use both (e.g., Table 1 and line 368 vs. lines 328, 334, 335). Maybe using always the same could be interesting.

Author Response

We thank the reviewer for his valuable and precious contributions. All points have been changed accordingly and exstensive editing of English have been made by native English-speaking colleague.

Reviewer 2 Report

Introduction

1.- L 37.- I think that authors should ellaborate in more detail the different ways to use the microballoons for liver embolization procedures. There may be the case in which the device is used for occluding an afferent tumoral vessel in a very superselective way. But also it may happen that the device is used in a non-occlussive manner with the aim of decreasing the distal pressure but still allowing the direct passage of blood between the balloon and the vessel wall. Undoubtely, but always in my personal opinion, this explanation will be very useful for the readers. It will give some light in the understanding of the application of these devices both in superselective and in lobar treatments for liver tumors.

2.- L42-45.- Previous investigations concerning the use of non-occlussive devices, as for example the “Surefire”,  allowed to know more about the pressure gradients while using sub-occlusive devices. I think that authors should add, at least, the following references (Rose SC, et al. “Downstream hepatic arterial blood pressure changes caused by deployment of the surefire antireflux expandable tip”. Cardiovasc Intervent Radiol. 2013;36:1262-1269. Rose SC, et al. “Quantification of Blood Pressure Changes in the Vascular Compartment When Using an Anti-Reflux Catheter during Chemoembolization versus Radioembolization: A Retrospective Case Series”. J Vasc Interv Radiol. 2017;28:103-110) and comment about them.

3.- L42-45.- Other authors have already published some in-vitro research concerning the use of microballoons. Maybe authors should considere its discussion (Aramburu J, et al. “Numerical zero-dimensional hepatic artery hemodynamics model for balloon-occluded transarterial chemoembolization”. Int J Numer Method Biomed Eng. 2018 Jul;34(7):e2983. doi: 10.1002/cnm.2983. Aramburu J, et al. “In  Vitro Model for Simulating Drug Delivery during Balloon-Occluded Transarterial Chemoembolization” Biology (Basel)2021 Dec 16;10(12):1341. doi: 10.3390/biology10121341.) because they try to explain which is the impact of MB in flow dynamics.

4.- L51.- Please add a reference.

5.- Has MBI any other application in non-hepatic territories? Please add some concise information, if this is the case, about it.

6.- L52.- According to the last BCLC recommendations (Reig M ref “2”) TARE is also an option for “early” and “very early” HCC. Maybe authors could re-elaborate this paragraph concerning the clinical indications of transarterial therapies in primmary and secondary liver tumors.

M&M

7.- L73 y 74.- Define b-TACE and b-SIRT.

8.- L96.- Please explain in what consists, practically, the evaluation of “the first 10% of the articles”.

Results

9.- L205.- Change “founding” to “finding”.

10.- L213-224.- Is it correct to refer as “assymptomatic abscess” or as “biloma” for the same/very similar clinical situation wich is the presence of an intrahepatic collection that has appeared in both cases after the manipulation?

Discussion

11.- L 362.- Which “routine folow-up” method was used?

12.- L363-368.- This is an important paragraph with a satisfactory “take home” information. It would be, however, very positive if authors could give more information, maybe in M&M (L89-93) or Results, concerning  “the different systems of evaluation that have negatively influenced”. Please add information or clarifie if it has already been detailed elsewhere.

13.- For educational purpuses maybe authors could add more technical  information; concerning, for example, the peculiarities of the four different devices that have been used (any preference for different scenarios?); also the technical recommendations for the use of MBI in superselective “occlussion” and the use of MBI in non-superselective “sub-occlussion”.

Author Response

Introduction

1.- L 37.- I think that authors should ellaborate in more detail the different ways to use the microballoons for liver embolization procedures. There may be the case in which the device is used for occluding an afferent tumoral vessel in a very superselective way. But also it may happen that the device is used in a non-occlussive manner with the aim of decreasing the distal pressure but still allowing the direct passage of blood between the balloon and the vessel wall. Undoubtedly, but always in my personal opinion, this explanation will be very useful for the readers. It will give some light in the understanding of the application of these devices both in superselective and in lobar treatments for liver tumors.

We thank the reviewer for raising this crucial point. The aim of this systematic review is to report published literature on  potential clinical use of the microballoon catheter. To date there’s no any available evidence demonstrating it’s employment in liver embolization procedure in a non occlusive manner. Is thus impossible to reflect such clinical practice in this typology of paper.

The article describe in detail all different reported modality of employment with regards to degree of catheterization  selectivity (superselective vs selective), number of feeders catheterized (SOFA technique) and technique (RAIB-TACE) and materials employed in embolization (particles, lipiodol).

2.- L42-45.- Previous investigations concerning the use of non-occlusive devices, as for example the “Surefire”,  allowed to know more about the pressure gradients while using sub-occlusive devices. I think that authors should add, at least, the following references (Rose SC, et al. “Downstream hepatic arterial blood pressure changes caused by deployment of the surefire antireflux expandable tip”. Cardiovasc Intervent Radiol. 2013;36:1262-1269. Rose SC, et al. “Quantification of Blood Pressure Changes in the Vascular Compartment When Using an Anti-Reflux Catheter during Chemoembolization versus Radioembolization: A Retrospective Case Series”. J Vasc Interv Radiol. 2017;28:103-110) and comment about them.

Thanks for focusing the attention on all antireflux devices, both the papers were cited in the introduction.

3.- L42-45.- Other authors have already published some in-vitro research concerning the use of microballoons. Maybe authors should considere its discussion (Aramburu J, et al. “Numerical zero-dimensional hepatic artery hemodynamics model for balloon-occluded transarterial chemoembolization”. Int J Numer Method Biomed Eng. 2018 Jul;34(7):e2983. doi: 10.1002/cnm.2983. Aramburu J, et al. “In  Vitro Model for Simulating Drug Delivery during Balloon-Occluded Transarterial Chemoembolization” Biology (Basel)2021 Dec 16;10(12):1341. doi: 10.3390/biology10121341.) because they try to explain which is the impact of MB in flow dynamics.

Thanks for raising this point. Although those studies are milestones in the understanding of blood flow redistribution, non-clinical studies (not involving humans) were excluded and both the studies are in vitro. 

4.- L51.- Please add a reference.

References have been added.

5.- Has MBI any other application in non-hepatic territories? Please add some concise information, if this is the case, about it.

Thanks for raising this point. The focus of the review is to analyze literature on liver malignancies embolization with MB. The title of the review was changed accordingly.

6.- L52.- According to the last BCLC recommendations (Reig M ref “2”) TARE is also an option for “early” and “very early” HCC. Maybe authors could re-elaborate this paragraph concerning the clinical indications of transarterial therapies in primmary and secondary liver tumors.

Thanks to the reviewer for this valuable suggestion. Text was changed accordingly.

M&M

7.- L73 y 74.- Define b-TACE and b-SIRT.

Text has been changed accordingly in the introduction section

8.- L96.- Please explain in what consists, practically, the evaluation of “the first 10% of the articles”.

We thank the reviewer. Text has been changed in order to make the process of reviewing more understandable to the reader.

Results

9.- L205.- Change “founding” to “finding”.

Changed accordingly

10.- L213-224.- Is it correct to refer as “assymptomatic abscess” or as “biloma” for the same/very similar clinical situation wich is the presence of an intrahepatic collection that has appeared in both cases after the manipulation?

Is beyond the potentiality of the review to reclassify  the Adverse event categorized by others’ literature. We suppose that the distinction was made radiologically relying on the absence or presence of air in the collection.

Discussion

11.- L 362.- Which “routine folow-up” method was used?

Thanks for raising this criticity. Text was changed accordingly.

12.- L363-368.- This is an important paragraph with a satisfactory “take home” information. It would be, however, very positive if authors could give more information, maybe in M&M (L89-93) or Results, concerning  “the different systems of evaluation that have negatively influenced”. Please add information or clarifie if it has already been detailed elsewhere.

We rephrased in “Oncological response, despite being negatively influenced by different systems of evaluation across countries (RECICL and mRECIST) that impeded a meta-analysis”. Thanks for the suggestion.

13.- For educational purpuses maybe authors could add more technical  information; concerning, for example, the peculiarities of the four different devices that have been used (any preference for different scenarios?); also the technical recommendations for the use of MBI in superselective “occlussion” and the use of MBI in non-superselective “sub-occlussion”.

We thank the reviewer for raising this point. Text was changed accordingly.

Reviewer 3 Report

This review article is well written, but there is only one issue to be corrected.  Microballoon is used not only for treatment of liver tumors but also for other interventions such as NBCA injection and coil placement.  Thus, it would be desirable to change the title of this paper as “Microballoon interventions for liver tumors: review of literature and future perspectives”.

Author Response

We thank the reviewer for the valuable advice. Title have been changed accordingly to reviewer's suggestion.